# Problematic Use of Nitrous Oxide by Young Moroccan–Dutch Adults

**DOI:** 10.3390/ijerph18115574

**Published:** 2021-05-23

**Authors:** Ton Nabben, Jelmer Weijs, Jan van Amsterdam

**Affiliations:** 1Urban Governance & Social Innovation, Amsterdam University of Applied Sciences, P.O. Box 2171, 1000 CD Amsterdam, The Netherlands; a.l.w.m.nabben2@hva.nl; 2Jellinek, Department High Care Detox, Vlaardingenlaan 5, 1059 GL Amsterdam, The Netherlands; Jelmer.Weijs@jellinek.nl; 3Amsterdam University Medical Center, Department of Psychiatry, University of Amsterdam, P.O. Box 22660, 1100 DD Amsterdam, The Netherlands

**Keywords:** laughing gas, N2O, problematic use, Moroccan–Dutch, Muslim, treatment demand, nitrous oxide, adverse effects, recreational use, drug dependence, migrants

## Abstract

The recreational use of nitrous oxide (N2O; laughing gas) has largely expanded in recent years. Although incidental use of nitrous oxide hardly causes any health damage, problematic or heavy use of nitrous oxide can lead to serious adverse effects. Amsterdam care centres noticed that Moroccan–Dutch young adults reported neurological symptoms, including severe paralysis, as a result of problematic nitrous oxide use. In this qualitative exploratory study, thirteen young adult Moroccan–Dutch excessive nitrous oxide users were interviewed. The determinants of problematic nitrous oxide use in this ethnic group are discussed, including their low treatment demand with respect to nitrous oxide abuse related medical–psychological problems. Motives for using nitrous oxide are to relieve boredom, to seek out relaxation with friends and to suppress psychosocial stress and negative thoughts. Other motives are depression, discrimination and conflict with friends or parents. The taboo culture surrounding substance use—mistrust, shame and macho culture—frustrates timely medical/psychological treatment of Moroccan–Dutch problematic nitrous oxide users. It is recommended to use influencers in media campaigns with the aim to decrease the risks of heavy nitrous oxide use and improve treatment access. Outreach youth workers can also play an important role in motivating socially isolated users to seek medical and or psychological help.

## 1. Introduction

Nitrous oxide (N2O; “hippy crack”; laughing gas) is a safe and widely used anaesthetic agent and is also used as a propellant for whipped cream. Due to the modest euphoric effects [1], nitrous oxide is also used recreationally [2,3]. Nitrous oxide abuse is emerging throughout the USA, Australia and Europe.

### 1.1. Prevalence of Nitrous Oxide Use

Recreational nitrous oxide use is defined as fewer than ten nitrous oxide balloons per event or per month [2]. According to the Global Drug Survey [4,5], nitrous oxide has become the seventh most used recreational drug in the UK with a last year use prevalence of 21% in 2013 (average age: 24.3 years). The 2016/17 Crime Survey for England and Wales reported that among the 16–24 year old subjects, the last year prevalence of nitrous oxide use was 9.3% (males, 11.1%; females, 7.4%) [6]. The Global Drug Survey 2016 reported that last year use in the Netherlands was 33% vs. 38% and 23.7% in the United Kingdom [4]. Indeed, in the Netherlands, nitrous oxide is gaining popularity in both secondary school students [7] and the general population [8], especially among young adults and those who are highly educated [9]. More than a quarter of the students (30%) of post-secondary vocational education reported that they had used nitrous oxide in 2019 with 8% in the past month [10].

Pupils with a non-Western migrant background have more experience with the use of nitrous oxide than pupils without a migrant background [9]. A recent survey among Dutch adolescents (14–18 years old) corroborates these results as nitrous oxide use was associated with a non-Dutch ethnic background (OR = 2.10), lower education levels (OR = 1.88) and binge drinking (OR = 2.49) [11]. Last year and last month use of nitrous oxide among Dutch people with a non-Western migrant background (3.6% and 1.2%, respectively) is higher than among those with a Dutch background (2.5% and 0.8%, respectively) [9].

The majority (77%) of nitrous oxide users were unaware of the drug’s harmful effects [5] and believed that the drug was safe because of its legal status [2]. However, the legal status for most young adult “party users” is irrelevant in their decision to use nitrous oxide [2]. Due to the increasing recreational use of nitrous oxide over the past ten years, including world-wide [12], a legal prohibition is being considered to counter this [13,14]. For the time being, nitrous oxide has been subject to the Commodities Act in the Netherlands (since July 2016). In France, some towns have already banned the sale of the gas to under 18-year-olds, and initiatives are underway to ban the sale to minors nationwide [15].

### 1.2. Adverse Effects of Nitrous Oxide Use

The frequency of nitrous oxide use varies with the largest group using nitrous oxide once or twice in the past year (45%), while 16% did so more than ten times [16]. Incidental use of nitrous oxide is hardly, or not at all, associated with major damage to health. Frequently reported side effects are headache, dizziness, fainting, accidental falls and tingling of hands and feet. Nitrous oxide induced persistent numbness/tinging (peripheral neuropathy; paraesthesia) in hands or feet is a dose-dependent neurological effect [5,17] with persistent numbness in 4.3% of last year users [5].

However, the health damage of nitrous oxide, including its dependence risk, increases rapidly with daily use in higher daily doses; occasionally hundreds of cartridges were inhaled per day [18]. Especially after the emergence of larger nitrous oxide tanks (2-Kg tanks) in 2017–2018, the number of serious neurological complaints increased [19]. Following intensive use (10–20 balloons or more daily), serious neurological complications such as myelopathy and generalized demyelinating polyneuropathy [20,21,22,23,24] can occur due to nitrous oxide induced vitamin B12 deficiency [3,25]. These complications are usually reversible and can generally be resolved through vitamin B12 supplementation (injections), but the neurological recovery may be retarded of even incomplete if patients continue to use nitrous oxide [26].

The Drug Incidents Monitor registered a limited but increasing number of incidents involving nitrous oxide (in 2018: 51 reports; 0.8% of all drug incidents) [9]. In 2019, the Dutch Poison Centre (NVIC) received 128 reports of health complaints in people aged 13 and over after recreational use of nitrous oxide. Neurological complaints were reported in about a third of the reports, which indicated (chronic) abuse of large quantities [18]. In 42 out of 78 hospitals contacted by the Dutch Neurologists’ Association, 64 young adults (mean age: 22 years) with a partial spinal cord injury following nitrous oxide use had been treated in the past two years (2018–2019) [27], but the actual number of victims is believed to be greater. The recent increase in nitrous oxide related traffic incidents is another point of concern [9].

### 1.3. Moroccan–Dutch Young Adults

Over the past three years, the number of nitrous oxide-related patients at the Reade rehabilitation centre, the emergency department (OLVG hospital, Amsterdam, The Netherlands) and addiction care (Jellinek) has steadily increased. Although this has not yet been explored and exact figures are not available, it involves several dozen young adult clients each year (mainly in their twenties), the majority of whom have a Moroccan migrant background. That is why this explorative study focused specifically on Moroccan–Dutch youth. In the next section, a brief overview is given of the belief, substance use, health and youth culture in this group, because these appeared to be determinant of nitrous oxide abuse.

A large part (94%) of Moroccan–Dutch, including those belonging to the second generation [28], consider themselves religious but non-practicing Muslims [29], while the religious perception of their parents is stronger than that of the young adults [30,31,32]. Particularly among young adult Muslims, Islamic beliefs often have a strong social influence in how they perceive mutual relationships and (online) social networks [33]. Compared to the Dutch, Moroccan–Dutch generally have more local and family ties, more homogeneous networks and their norms and values more conservative. For the Moroccan–Dutch, religion plays a greater role in life, tolerance toward “modern” behaviour is more limited and they are also less satisfied with their lives than the Dutch [34].

### 1.4. Drug Use in a Taboo Culture

Ethnic minorities, like the Moroccan–Dutch community, live relatively more often in socially deprived areas and are more often less educated and unemployed; factors associated with problematic alcohol and drug use [35]. Perhaps even more relevant with respect to alcohol and drug abuse by Muslims is that it is part of Muslim taboo culture: the use of substances is “haram” (sinful). As such, young adult Muslims may find it difficult to regulate their drinking behaviour, which can lead to excessive alcohol consumption, especially in problematic home situations (particularly in residential youth care) [36]. Compared to Turkish–Dutch, Moroccan–Dutch drink 1.7 times more alcohol and three times more excessive alcohol (15.4% and 5.3%, respectively) [37]. Moroccan–Dutch in Amsterdam have an increased risk of alcohol abuse (OR = 2.25; 95% CI: 1.44–3.53) [38] and regular drinkers of Moroccan descent have a significantly higher risk of developing “binge” drinking and alcohol dependence than regular drinkers in other ethnic minority groups [39]. With respect to cannabis, problematic use is more common among users with a non-Western migrant background than their Dutch peers (33% and 17%, respectively) [9].

Problematic use of drugs may also arise from various personal, sociocultural factors (poor social support from family or friends) and/or psychological factors (low self-esteem, marginalization, discrimination, stigmatization, loneliness, boredom) [40]. The low self-esteem may also result from internalizing problems related to poor parental support, parent-child conflict, school/peer (being bored) and the adolescent’s perceived discrimination, as observed among immigrant adolescents in the Netherlands [41]. Young Moroccan adults in the Netherlands often feel discriminated against mainly on the basis of their religion and Muslim identity [42] and do not count as fully fledged [43], and feel stigmatized, discriminated and excluded [42]. A recent systematic review showed that discrimination was associated with heavy and hazardous drinking, particularly within stress and coping frameworks [44]. Furthermore, stigmatization of drug dependence can hinder people’s search for treatment [45,46]. Avoiding seeking help and/or denying a (serious) emotional problem can also escalate problems. That fact that psychiatric illnesses are often taboo within Moroccan families is another point of concern [47]. Indeed, young adults with a Moroccan and Turkish background are underrepresented in regular youth mental health care: about half of what can be expected demographically [48,49].

In this qualitative exploratory study, we had the unique opportunity to interview young Moroccan–Dutch heavy users of nitrous oxide with the aim of identifying determinants of problematic nitrous oxide use. This is unique because this group is reluctant to share their thoughts about substance use, which for them is intertwined with shame.

## 2. Methods

It is difficult to obtain access to problematic nitrous oxide users outside of institutional settings. This problem is particularly acute for nitrous oxide as these users rarely come into contact with treatment services and those that do are unlikely to be representative of the larger population of users. The recruitment procedure did not produce a statistically random sample, mainly due to heavy impediment by feelings of shame among this group of problematic nitrous oxide users. Based on earlier signals in the research field, we opted to perform our research within the New-West district of Amsterdam, where relatively many Moroccan–Dutch people live [2,50]. In this neighbourhood mainly Moroccan–Dutch young adults show problematic nitrous oxide use, which was our primary inclusion criterion.

The group of problematic nitrous oxide users consisted of two groups: seven young adults (5 men, 2 women; 19–28 years) who were in clinical or outpatient treatment (Reade and Jellinek) for their problematic use of nitrous oxide and six young adults (5 men, 1 woman; 19–22 years) who heavily use nitrous oxide, but who are not in treatment. The users who were not in treatment (N = 6) were recruited by youth workers in the New-West district who applied the snow-balling technique. The other group (N = 7) were recruited by patients’ practitioner during the intake in the clinical and outpatient treatment centres. All interviewees had a Moroccan migrant background (aged 18–29 years) and lived with their parents/sister.

A list of topics were drawn up, covering questions about their use profile (frequency, dose), motives of use, awareness about health risks, effects encountered and their response to them and determinants of seeking help or not. This list (cf. Table 1) served as a guideline for the (in-depth) interview. Prior to the in-depth interview, the seven participants who were in treatment were interviewed for one hour about issues, such as upbringing, faith, family, friends, education, work, going out and their expectations for the future. The in-depth interview of approximately 2.5 h was mainly focused on their problems with nitrous oxide and additional questions were asked about motives for use, frequency and dosage, health complaints and use of social media. Furthermore, information was collected about the (taboo) culture around alcohol and drug use, their perception of health risks and care provision and why they entered health care at such a late phase. The six young people who did not receive treatment were interviewed for approximately 1.5 h with questions that were based on a shorter list of topics. All respondents insisted that the interviews not be recorded. The participants received a EUR 50 voucher as compensation for the interview. The professionals were interviewed for around 1.5 h about the living environment, the social domain of neighbourhood youth from different ethnic backgrounds and the risk factors to which they are exposed.

## 3. Results

The problematic use of nitrous oxide was a sensitive subject for all young adults; they were not eager to speak about it and some were suspicious or ashamed to discuss their customs. They did state that through their stories they hoped to make young adults more aware of the health risks that nitrous oxide can cause. The young adults interviewed are cultural Muslims, where faith is important, but they do not practice their faith (no praying or visiting the mosque). Some are selective Muslims who regularly participate in the social and ritual practices, though not very frequently. Some respondents highlighted the double standard in attitude toward drug use; that it is more easily overlooked and more accepted for men to use drugs than women.

### 3.1. Starting with Nitrous Oxide Use

In their spare time, the young adults are often elsewhere other than at home as the usually small houses in which they live afford little privacy. There is relatively little family income as the fathers often have low-skilled work, while the mothers mainly do the housework. The young adults themselves follow or followed vocational training and started working after having obtained their professional diploma. A few have dropped out halfway through and are unemployed. Some said they have a generous income for their age; some young people were vague about how they earned money but did hint that they are involved in illegal sales (nitrous oxide and cocaine) and profit well from it. One respondent (male, 22 years old), who is also selling nitrous oxide, has young people from the neighbourhood who deliver nitrous oxide tanks to customers for him. This picture is consistent with the stories of respondents, of whom at least half report they have accumulated debt (including health insurance, student debt or debts to family members) as a result of excessive nitrous oxide use. Finally, after having spoken with youth and social workers in the field, it would be easy to get the impression that nitrous oxide is popular only with marginalized youth in street cultures. However, other young adults who also reside in these neighbourhoods never had contact with youth work or the police, have completed an education and found work. The conversations show that they encountered nitrous oxide via other routes (friends, colleagues at work or in the nightlife circuit).

The respondents’ curiosity was often triggered by the “hype” around nitrous oxide. Cool artists started singing positively about balloons in clips [2], some friends had nitrous oxide whippets (small steel cartridges of 10 cc), girls only wanted to come to your party when there was nitrous oxide, local shops started to advertise and friends started trading with nitrous oxide via social media. Most young people started using nitrous oxide when they were underage (15–17 years old). In all cases, this was done in groups: during a school trip, after school at someone’s home, together with girls and other friends in a hotel room, a shisha lounge, garage, park, a beach tent or on vacation in Morocco, as a respondent still remembered well. “My first time was in an underground bar in Tangier where we used balloons together with some friends. Nitrous oxide is not allowed in Morocco, but someone from the Netherlands had smuggled in a cargo load and made sure that customs would turn a blind eye. There was unlimited nitrous oxide at that party, and I used a lot of balloons for the first time. The waiters were all the time filling balloons through a whipped cream syringe. I didn’t have to pay anything because us girls never have to pay for their drinks and balloons. We think that is normal. That’s the way it goes” (Woman, 27 years old).

One male respondent (19 years old) said that the use of nitrous oxide balloons started when he was bored with friends. The atmosphere is immediately more fun and there are quite a lot of guys in the area who do it. Therefore, it is not that special. Although he saw people tripping out. Young people who have nothing to do with it walk away automatically when it is balloon time. One respondent remembers the time when he was in a park with a group. One friend suddenly showed up with a box of nitrous oxide cartridges. He was a novice and had never drunk, smoked or drank alcohol before. “I didn’t want to at first, but when I saw that it made them happy, I also wanted to try a balloon. I felt tingling in my head. It was a wonderful feeling” (Man, 19 years old). A 17-year-old woman received her first nitrous oxide balloon from a gas tank. During that period, she spent a lot of time with a friend whose parents were often away in Dubai. “There was usually nitrous oxide, a 2-Kg tank you know. We were four girls. They already had experience with balloons. The first time I only did four. It was a nice, floating feeling” (Woman, 19 years old). When school closed (due to COVID-19) she started using almost every day with a girlfriend where boys always came with new tanks. “I never had to pay for it. It was always there.”

### 3.2. Drug Expenditures

The idea that nitrous oxide is cheap may apply to recreational users who occasionally use balloons, but not to problematic users who think in terms of 2-Kg tanks costing EUR 40–50 (for about 125–150 balloons) and meanwhile have used hundreds of such tanks or spent EUR 30,000–50,000. One respondent still has a debt of EUR 11,000 to his sister, while another said: “I inhaled nitrous oxide worth 30,000 euros, resulting in a spinal cord injury.” In contrast, the women interviewed said when they are going out, they rarely have to pay for their entrance, alcohol or nitrous oxide balloons, which in turn sometimes leads to psychological pressure for some of them when the “generous” giver starts making allusions about getting something in return. Conversations with women, stories circulate, confirm previous findings in Amsterdam [51], about so-called “balloon hookers” who have sex with young men in exchange for free nitrous oxide; these boys are referred to by them as “Jacks”. Some women criticize boys’ double standards. They complain that girls expect them to have to “arrange” the nitrous oxide, but at the same time, partying without girls is less fun. “So, if you want to slow down the use of nitrous oxide in boys, you have to convince the girls not to go to those nitrous oxide parties”, said a woman (18 years old).

### 3.3. Perception and Experience with Alcohol and Drugs

All participants reported that from a religious point of view, alcohol and drug use is labelled very negative (haram) in their family, and that this taboo subject is hardly ever discussed at home. With the exception of one respondent (Woman, 19 years old) whose father is no longer a Muslim (mother is). “The mosque warns about the errors of mind caused by drug use. It takes you from following the path. You get psychologically entangled and it causes confusion” (Man, 29 years old). According to one respondent, drug use is not necessarily discouraged for health reasons, but because the Holy Quran simply rejects the use of intoxicants. “I’ve always been told it’s forbidden in my religion, otherwise my parents might use it too” (Man, 22 years old). Another man said that his mother once made a comparison with glue sniffers in Morocco and that young people in the Netherlands would suffer the same fate as the poor street children in Morocco. One respondent (Man, 19 years old) said that he was warned about drugs from an early age: “You shouldn’t do anything that destroys your body”. When he was 13 years old, his older brother impressed on him that he should never accept a joint from someone else, because then he would no longer be clear-headed when praying. It is also striking that most adolescents in secondary school (including secondary vocational education) said that they had never been taught about alcohol and drugs. It may even happen that the father uses cannabis, but forbids his children to use it, as two women have experienced. “Drugs are a no go with us believers because they destroy your body. But apparently this rule does not apply to men. My father smokes hash every day. That’s a double standard” (Woman, 27 years old).

The respondents often reported that they do not smoke at home or drink alcohol out of respect for their parents. Yet, many respondents have used alcohol and cannabis. A minority experienced MDMA or cocaine at some point but prefer to keep it secret to avoid conflicts at home, and devise excuses and strategies to remain out of conflicts. As young people get older, they discover the nightlife where people drink and use drugs, so the likelihood of conflicts increases. In the stage of adolescence, they start to make their own plans, whether drugs are prohibited or not. Whoever comes home with a tobacco or alcohol smell can expect the wrath of the father of the house. “Nitrous oxide is legal, and your parents are not aware because it is odourless, and the effects are not visible. Maybe it will lessen the pain of your lie to your parents a bit” (Man, 26 years old).

Some say they do not follow the rules and opt for confrontation and return home drunk or stoned. “My parents’ sermons may have helped in the beginning, but as you get older, it makes you shit” (Man, 19 years old). A woman (18 years old) no longer wants to act hypocritically and has told her parents that she smokes, sometimes drinks and uses nitrous oxide. Respecting their parents’ restrictions, some arrive at home after the parents are asleep or decide to sleep elsewhere. A respondent who started to discover the nightlife stayed over with her Dutch girlfriend in order not to embarrass her parents and to maintain her own sovereignty. Another respondent preferred to spend the night with a “neighbourhood junkie” or in his own car in an abandoned parking space. He told us that there’s no point in talking about it because it is haram and simply not allowed. They know that he sometimes drinks alcohol. “Once I came home drunk at 5.30 in the morning. I was late and I had drunk a lot and got into a huge dispute with my father. As punishment, I had to stay indoors for a few days. So out of respect for my parents, I prefer to stay away when I’ve been drinking” (Man, 21 years old).

While their parents are often strongly against drugs, the respondents’ friends are more tolerant. One respondent (Woman, 19 years old) told how she started experimenting with cannabis within a group of friends between breaks and after school. Young people also talk about parties in shisha lounges, hotel rooms, clubs and private homes arranged by landlords with connections to nitrous oxide sellers. Interacting with friends of different ethnicities in your spare time can facilitate acculturation, so that these young people become more tolerant of drugs. On respondent (Man 22, years old) says that he did not drink much alcohol. His last time was six months ago on New Year’s Eve. However, he also realised that “the flesh is weak” once he starts drinking. “We then order large bottles of vodka that we are going to wave around (laughs). By buying a bottle you show that you got it right. Even if you only have 50 euros, you still buy that bottle. You understand?” He finds it difficult not to drink because he lives in the Netherlands where alcohol is so easy to get. If the rules at home are so strict then he’s even more driven to disobey. The Dutch are tolerant and more accepting of alcohol use. Moroccans, however, do not learn how to drink from peers or family members and so once they start drinking cannot limit their alcohol intake he says. “When we drink and do balloons, it is often out of a problem and not for fun” (Man, 21 years old).

Liking something, but also feeling guilty and bad at the same time leads to confusion, tension and doubt about who you are and who you want to be. The participants reported that they must always “switch” from parent culture to school and street culture, and vice versa, and each time play a different role. “It takes a lot of energy if you have to turn that button on and off all the time” said one respondent. (Female, 27 years old). Another woman (19 years old) has been told several times by her father that she is a whore if she does not come home on time. Tensions sometimes run so high that she longs for a place elsewhere. Young people often feel misunderstood by their parents and cannot be themselves in the stressful, masculine street culture. In street culture, boys collide again with “feminine” school culture which stimulates self-expression and self-development. A “mismatch” between teachers and students can be disastrous for school performance and thus perpetuate a negative self-image. According to some young people, this has a large impact on the way they live their life. Due to religious (Moroccan) upbringing, which discourages substance use, Moroccan–Dutch young adults use alcohol and drugs less often than their native Dutch peers. Likewise, the young adults interviewed also seldom drink alcohol, with an occasional outlier for some. Cannabis is more commonly used, especially in street cultures, while cocaine and ecstasy are rarely used, with a few exceptions. However, nitrous oxide is an exception that apparently confirms the rule.

### 3.4. Motives for Nitrous Oxide Use

The recreational use of nitrous oxide (for example holiday, birthday, nightlife), turned out to be more problematic among the respondents who struggle most with their Moroccan culture in Dutch society and suffer from an identity crisis. Reported motives (set factors according to the Zinberg model; cf. Discussion) to use nitrous oxide include to avoid boredom, seek relaxation, reduce stress and to stop compulsive negative thoughts. The pressure of working, studying, maintaining friendships and lying to your parents is stressful. Nitrous oxide can provide comfort through temporary relaxation and relief and seems to offer the only respite. One respondent (Man, 19 years old) said: “I was so busy in my head that I could only relax with balloons during the weekend.” Others mention emptiness, loneliness, depression, discrimination, conflicts with friends or parents, a loss of recognition and appreciation, an inability to express feelings and fear of failure. “You always get the feeling that you are a second-class citizen,” a respondent (Man, 26 years old) told us. He said that even his Dutch mother-in-law does not trust him because he is a Muslim. A 19-year-old-woman tells us that balloons gave her a happy feeling in the beginning, but she slowly realised that it was also a flight from problems. The more she started using the more she suffered from amnesia and concentration problems. “I kept saying to myself: this is my last balloon, this is the last balloon, this is…. Nitrous oxide makes everything shit. It makes you reckless. But what did I have to lose?” Some respondents even noticed other users surpassing their own use during holiday periods and the COVID-19 crisis. It is a dangerous time, they said, because young people do not know where they stand and how long it will take before they can go back to school, play sports and go on holidays. Boredom set in. “Corona made the problem even worse. I got into a crisis about who I was. I am Moroccan yes, but I do feel stuck between two worlds. You don’t really do well with anyone. That’s why I wanted to step out of reality” (Woman, 19 years old). Another woman (27 years old) says that she can stay in control with balloons. One advantage is that parents do not notice. “They don’t see it in you, it doesn’t smell like alcohol or tobacco. You can just keep talking. They don’t notice your pupils. Nobody notices. And that’s why it’s so dangerous, because it’s attractive. But at the same time, you feel guilty.”

### 3.5. The Emergence of Problematic Use

With the arrival of the large 2-Kg nitrous oxide tanks on the drug market (only EUR 40) around 2017–2018, the use of nitrous oxide increased rapidly. Young entrepreneurs—they were also considerable users themselves—started a lucrative business focused on the increasing demand from their own neighbourhoods. Delivery services worked 24/7 so that orders could also be filled at night. Nitrous oxide is probably the first drug (except for cannabis) that Moroccan–Dutch people sell to each other, which may explain the rapid spread in the residential areas of New-West. Some youngsters started to discover the regular nightlife of clubs, shisha lounges and late after-parties and in parallel, hotel parties where friends could gather using nitrous oxide every weekend. “Everything was innocent up to the point of those big tanks,” one respondent (29 years old) told us. He could now pump large balloons one after the other and felt like he was “trapped in a cycle of use without interruption and high all the time.”

Two women reported that they regularly ended up in private houses (10–15 people) where 10-Kg tanks were delivered non-stop all weekend. All respondents started using more when they switched to the larger tanks, which were easier to operate and, according to some, were also safer to use. “If you can get it that easy, it can’t be harmful, can it? Surely the government would have acted long ago and banned the drug?”, one respondent declared (Man 26 years old). The use of Nitrous sped up immensely with the arrival of 2-Kg tanks on the market. Not least because of ease of use. “You open that tap and you can blow up as many balloons as you want”, one user (Man, 22 years old) says. He laughs about the fuss he had before with those old- fashioned whippets. “Each time putting a whippet in the holder, turning it on, squeezing gas into the syringe, balloon over the valve and blow it up. Time after time and very tiring afterwards.” With a tank of 2 Kg (125–150 balloons) he goes faster and could also order a new one faster. Another respondent talks about his balloon friends. In the beginning they would start with one tank, but that soon would become two or three at a time. Nobody thought it was addictive because they did not use every day. “It was just fun. And it drives girls crazy and turned on. We put a lot of gas in hotel rooms. I paid the nightly rate of 120 euros. We smuggled the tank into the backpack. The two of us went in and then one went back to pick up two others” (Man, 21 years old).

However, not everyone reacted in a relaxed way to nitrous oxide. “Nitrous oxide was a real hype. Things started to get out of hand as it was used in larger groups. Emotions often ran high, there were arguments while others were tripping on that gas. They saw things that were not there. You discover that you prefer to be alone or in a small group without distraction” (Woman, 27).

After the “experimental phase”, use is increased in phases in the “party phase”. After an exuberant period, the original group slowly disintegrates and shrinks to a small core of users that keeps going and will not stop. Users who eventually end up in the peak phase become increasingly isolated and “binge” on their own at least three times a week on two to three 2-Kg tanks per session, or one 2-Kg tank almost daily. The maximum someone used was about eight tanks in a day-long binge. One likes large balloons (skippy balls), while the other prefers to inhale smaller balloons (one per minute). None of the young people interviewed said they had a limit. Simply because they quickly lose count and continue to use “by feeling”. Some enter a timeless vacuum during their nitrous oxide binge where hours and days seem to merge into one long trip. “You fall asleep with a balloon in your mouth, wake up after an hour and a half, and go on with balloons” (Man, 22 years old).

At this stage, most did not realize that health risks were increasing significantly. “I did hear stories about people getting paralyzed, but also saw that everyone around me kept doing it. If they can handle that much, then my body can handle even more”, boasted one respondent (male, 20 years old). The respondents gave various reasons for switching to problematic use. A few respondents said that the nebulous world of nitrous oxide provides temporary relief. You do not have to think about your pain and problems that you cannot solve anyway. One respondent (Man, 29 years old) says he started using more nitrous oxide a few years ago for more relaxation. However, the underlying reason was that he had a “burnout” due to a high workload and was stuck because of the conflicts at home (still lives with his parents). “When I was stressed out, I would go outside to use nitrous oxide as a flight. My head felt so full that it felt like relief at first.” However, at a certain point he could no longer switch properly between his work and his nitrous habit at night. One respondent (women, 27 years old) intensified nitrous oxide use following a traumatic event (intimidated and threatened by two unknown men). She felt distraught and started using a lot to muffle her thoughts and switched to tanks. She knew enough dealers. As she no longer trusted anyone, she started using it every day for five months; alone in the car on the outskirts of Amsterdam with a 2-Kg tank that took her six hours. “For me it was eat, work, call the dealer, use it and sleep. Every day for five months.” The dealers were happy with her because she also introduced other customers so that she sometimes got a tank for free. Another respondent (Man, 26 years old) was naive about the criminal double life of his friends. When the police searched for him, he went into hiding for two years and found himself in a shabby world of private parties, after parties, shisha lounges and temporary shelters. He started using nitrous oxide more and more, and in higher doses to reduce the stress that had arisen. There are also young people who gradually started using nitrous oxide more and more. One respondent (Man, 19 years old) worked in sales and afterwards went on a nitrous oxide tour with a few colleagues. He said: “We started with 2 Kg (75 balloons each), but that slowly increased to 8–10 Kg. We went on until 3 a.m.–4 a.m., when the tank was empty. You are building up tolerance, you know, which makes you want to use more and more each time.” In the beginning he always used with friends on weekends but ended up alone, along with his tank in his car nearby a parking lot. “I just wanted to space, so you can keep on rumbling. You wanted to stay in it. No, I had no limit.”

Due to increasing tolerance (habituation), the extent of the use of nitrous oxide can quickly increase, which can lead to problematic or excessive use. “The first balloons always feel good; then the horror starts”, one respondent said (Man, 21 years old). When the Netherlands went into “lockdown” due to COVID-19, he started to “build up rhythm” and he “shot” three 2-Kg tanks three times a week and recovered in between. A year ago, he was the jolly boy who arranged tanks, hotel rooms and girls eager to throw a party. However, he changed character and started to show unpredictable behaviour; quickly became angry and had fits of rage when girls wanted to go home. “I was lonely, had dropped out of school and the tank became my best friend. I slept in the car in garages.”

### 3.6. Adverse Health Effects of Problematic Use

The first signs of physical damage due to nitrous oxide use are tingling in your toes and feet, losing balance more quickly and falls become more common. Other symptoms are vascular complications and infarcts, feeling bloated and nauseated, heartburn, pain in the neck, the upper arms and shoulders and one’s physical strength can “completely” disappear within six months. Frostbite injuries have occurred [52,53], because the tank becomes ice cold during draining, which is not noticed due to the anaesthetic effect of nitrous oxide. Some users get blisters in the mouth, and on the tongue, arms and legs. Some of the wounds are very deep and healing is problematic, requiring multiple surgeries and resulting in scars with poor quality [52]. “Holding a pen was no longer easy. Writing became more difficult. My words were also worse” (Woman, 27 years old). The physical complaints are not permanent and (may sometimes) disappear. Many users apparently do not yet realize the relation between their excessive use and increasing physical discomfort, ending them up in the danger zone. “My desire for nitrous oxide was stronger than my mind”, said one respondent (Man, 19 years old). Until he could not get out of bed because his legs were paralysed. He was rushed to hospital by ambulance. After a lecture from the doctor and vitamin B12 injections, he was back on his feet a few weeks later. Once at home, he started using again and was readmitted after a few months. Due to their excessive use, some respondents became (temporarily) paralyzed in their legs, but largely recovered after intensive therapy. One respondent (Man, 26 years old) still cannot comprehend why he did not hear alarm bells when his neighbour ended up in a wheelchair.

Problematic users of nitrous oxide also reported mental problems. The respondents reported high tension, quarrels and sometimes violence between users or toward friends, partners and parents. Heavy users can become unpredictable and paranoid delusions can lead to violent behaviour. During our fieldwork, we had to carefully approach the cars with users in them, because they are in their self-created time capsule. A rough disturbance can lead to aggressive behaviour. One respondent (Man, 29 years old) felt he had become absent and absorbed in his self, after using nitrous oxide. Heavy users can suddenly become aggressive he warned. “Never knock on a car window where someone is spacing a tank, because you can get in trouble.”

Complaints reported by respondents during their binge of hundreds of balloons at a time and often more than a thousand a week are mood swings, anxiety attacks, paranoia and sometimes suicidal thoughts. Heavy nitrous oxide use reinforces negative emotions: high doses of nitrous oxide can lead to delusions, hallucinations and a feeling of depersonalization. “You’re going to talk to your tank and see things that aren’t there” one respondent (Man, 29 years old) told. He became so paranoid that he wanted to drive away to flee the evil spirits. At another time, he thought the girl, with whom he often used nitrous oxide, was trying to trap him. One respondent (Man, 21 years old) said that he makes things bigger in his head. When he uses alone in the car, he hears different voices that can be very disconcerting. He got scared and paranoid that someone was going to hurt him. One time he freaked out and put a gun to his temple. “In the space, you see bizarre things. I saw people turn to black smoke. If you are a believer, you get anxious when the devil disguises himself in delusions” (Man, 26 years old). Flashbacks also occur. One respondent (Woman, 27 years old) described reliving bad memories of her father in childhood; nitrous oxide creates delusions that take you to an unknown underworld. Another respondent got into a faith and identity crisis, because nitrous oxide turned his thinking and belief world upside down. He sought help from a kind of imam, who advised him to stop using nitrous oxide; you will then automatically return to the pure path. “As a Muslim you have to be strong because you know that you are committing a sin if you use drugs or nitrous oxide. Many boys start to think about their beliefs when they are under the influence and get into conflict with themselves. That is difficult to explain. It’s very confusing and you can’t really talk to anyone about it. You must solve it yourself, but you don’t know how. That creates a lot of extra tension” (Man, 26 years old).

### 3.7. Social and Financial Problems

The original pleasure of recreational nitrous oxide may gradually develop into chronic use that is linked to physical, mental and social complaints. During the interview, some respondents realized for the first time that they may have occasionally been psychotic or had experienced hallucinations. The heavy users gradually isolate themselves, because of conflicts in the larger group. An example is in the story that heavy users often end up in a car somewhere in an abandoned place, along with his tank, a bag of balloons and a phone to call his dealer when the tank is almost empty. “One wants music, the other doesn’t; he wants to hear this, the other that again. Conversations sometimes disturb you. Everyone wants the best situation for themselves. And that creates tension” (Man, 19 years old). Users fall into social isolation: dropped out from school, got fired and got angry with that last loved one who cares about you. “You lose everything, your girlfriend, your friends, school, and I was evicted from home. You live in a bubble. It’s all about balloons” (Man, 21 years old). Debt accumulation is also a problem. You start buying tanks on credit from dealers who get exceedingly angry if you do not pay on time, but you desperately need a new tank. “While I was using, Nitrous was the only thing I could think of. I kept promising my dealer that I would get him the money quickly (1500-euro debt), even when I didn’t have it anymore. I kept making excuses and misusing his trust. The money will come tomorrow! I got back on credit for 10 Kg. He kept giving it to me and I kept delaying paying” (Man, 19 years old). His dealer got fed up with his stories and started threatening him, but his sister, to whom he already owed EUR 11,000, saved him that night in the nick of time.

### 3.8. Limitations on Access to Treatment

Heavy users mostly deny the problems associated with problematic nitrous oxide use by persisting in their belief that nitrous oxide is harmless. In addition, they hardly appear to discuss such problems with others, and initially downplay their use, risks and personal problems. As such, their demand for help is minimal because the problems are insufficiently recognized by their direct environment. It was already clear that substance use, and more specifically nitrous oxide use, are taboo subjects about which young people in the parent culture cannot openly exchange ideas. Most respondents receive little help and must solve their nitrous oxide problems themselves. Some respondents refer to their culture where it is often about pride and honour and that you do not want to damage your reputation. That is why users prefer to keep those thoughts (problems) to themselves. Some therefore speak of a “culture of silence” in which you keep your mouth shut about others. In street culture, breaking the silence to the authorities is often criticized as “snitching”. Many respondents said that they do not like to talk to their parents about their feelings, emotions, and fears, even if they would understand. Some feel that their parents’ ability to understand is limited so it would only create extra hassle. They are regularly told from home “not to go astray”. One respondent (Man, 19 years old) said that his mother told him not to hide if he was going to try things out; he could always talk to her about it. “And yet I found it difficult to talk about nitrous oxide. I was also ashamed of myself and felt that I was not ready for help.” Another respondent (Woman, 19 years old) said that she does not talk about it at home, because she is afraid of disappointing her parents. “My parents think I am a child prodigy. And then I’m going to tell you that I do balloons every day? Exactly what my father always warned me against. I don’t even tell friends. They have a prejudice that I am sitting here.” Another respondent (Man, 19 years old) also feels guilty toward his parents now that he is in Reade rehabilitation centre for treatment. He does not dare to face them and is happy that his older brother is a mediator. He feels defeated for hurting their pride for him. He feels very guilty about that and feels like he has failed.

Several respondents said that mothers, sisters or aunts often show the most compassion and offer the solution; fathers prefer to be “kept out of the loop” as much as possible. They contacted the doctor or psychiatrist, e.g., for the referral. It is often difficult for boys to talk about their feelings with other boys. They are in a world where it’s all about bravado, success, toughness, looking good, impressing girls, showing that you have balls. Aside from the masculine excuse, do not bother others with “your shit” because they are often in a mess themselves. “You have your pride. You shut up, because before you know it, gossip is going on around that you have problems” (Man, 26 years old). Indeed, Moroccan boys are reluctant to participate in group treatment offered by addiction care, where they must talk about their emotions; they are not at all used to speaking openly about their problems, so they often quit. On the other hand, several respondents have been approached by concerned friends about their extreme nitrous oxide use; some even dozens of times. However, due to increasing isolation, the outside world is getting increasingly less aware of it. It is significant that some respondents can only discuss their problem with an unknown or “safe” person who is understanding, not moralistic and unaware about their circle of friends. Somebody who is not threating and does not think you are a bad Muslim.

### 3.9. Counselling by Muslim Clergy and Professionals

All respondents said they were unable or unwilling to share their problems with their parents. They also did not want to bother their friends, because they were struggling with the same issues. The refusal of almost all respondents to consult a professional (youth or social worker in the field) or social worker is motivated by pride, shame or denial of the problem. Moreover, they do not know how aid is organized and are sometimes afraid of ending up “in a system”. In short, most problematic users keep muddling on and said that they do not have a good relationship with professionals or do not find them credible, because they do not understand the Moroccan culture well.

The aid to the respondents appears to follow two tracks: first the Moroccan approach (primarily from the point of view of faith) followed by aid based on the Dutch approach (psychology, medical). In accordance with the advice of parents, friends or family members, a Moroccan spiritual counsellor or care provider is first contacted, with whom psychological problems related to the use of nitrous oxide can be discussed. One respondent (Man, 21 years old) spoke about an imam who read the verses of the devil from the Holy Quran to cast out the devil. Another respondent (Man, 29 years old) first went to Morocco for a nitrous oxide detox to find “inner peace”. After his return, however, he quickly fell back into heavy use, after which he was admitted to Reade for a spinal cord injury.

One respondent (Man, 29 years old) first went to Morocco for help and then sought help in Amsterdam. “They (the Jellinek addiction facility) really helped me and gave me an insight into how it works for me when I crave nitrous oxide. It helped me get to know my behaviour better.” He said that this knowledge also helps him have better conversations with friends and to try to help them. “I tell them that they don’t have to be so afraid of Dutch aid, because it helped me a lot. I went to Morocco for an Islamic cure. Many verses from the Holy Quran are then read to expel the shaytaan (Satan). I felt that I was coming to terms again. I couldn’t have done that here. You forget what you’re doing here and break your daily routines” (Man, 29 years old).

In addition to the “fun” side about nitrous oxide on social media [2], there are already vlogs on YouTube in which imams express their concerns about nitrous oxide. It is therefore worth considering using influencers (role models) in campaigns, aimed at the risks and treatment of heavy nitrous oxide use.

## 4. Discussion

The number of nitrous oxide related health incidents has steadily emerged in the past three years and led to increasing concern at rehabilitation centres, the emergency department and addiction care in Amsterdam. In addition to the emerging adverse health effects, a steadily growing number of traffic accidents, where the driver was under the influence of nitrous oxide, have been reported. A growing group of mainly young migrants (a few dozen per year), in particular Moroccan–Dutch youth, have developed excessive use of nitrous oxide. Therefore, the city council aims to take extra measures (campaign online influencers, outreach work) among groups of frequent (problematic) users who are unaware of the risks of nitrous oxide.

### 4.1. Nitrous Oxide Use in a Taboo Culture

The first question of the inquiry was about the arguments why subjects wished to refrain from nitrous oxide use. All users mentioned the great taboo on alcohol and substance use in their religion and parent culture. Parents often have little or no experience with substance use and have therefore a limited frame of reference. Drugs are haram (prohibited) and parents want their children to feel the same way. The current generation of young people partly picks up this message (impeding factor), but there is also a substantial group that starts to experiment in the adolescent phase because they come to settings (going out, hanging out, juvenile detention, vacation, etc.), where the taboo on substance use is more or less ignored or deliberately violated. But what motives do young people have for experiencing nitrous oxide in a taboo culture? Young people who are in this process of acculturation feel a certain tension because the violation of rules and the disinhibitory behaviour are at odds with the guidelines of the faith that disapproves of or forbids the consumption of intoxicants. This abstinence paradigm, where there seems to be no middle way (self-regulation) between total abstinence or addiction, is reminiscent of the moral panic surrounding young drug users in often strictly religious Dutch fishing villages.

Except for a small group of problematic users (mainly Moroccan and Turkish youth), most recreational nitrous oxide users do not encounter drug related problems. Drug users may, however, encounter drug related problems, if set and setting are in disbalance. The set and setting are explained in the model of Zinberg [54]. According to this model (1) demographic and social-economic characteristics of the user, the motives, perception, previous experience with other drugs and the personal attitude towards the substance (“the set”) and (2) the social and physical environment in which drug use takes place (“the setting”) are decisive for the positive or negative experience of a substance. Becker [55] previously argued that controlled drug use is the result of a learning process in the peer group. Finally, it may have been relevant for the specific group investigated here, that drug use can also be an escape from the everyday banal routine or a reaction to an instable and rapidly changing world in which in the absence of sufficient social control, the urge to excess and kicks is magnified (e.g., [56,57,58]).

The fact is that young people experience an inner conflict and, out of a certain respect for their parents, start to feel more guilty. Therefore, young people experimenting with drugs prefer to do so outside of their neighbourhood to avoid gossip and social control. They must constantly come up with excuses and tricks to conceal the use of alcohol, cannabis or other substances (smell, behaviour, being under the influence, etc.). As such, nitrous oxide provides an ideal alibi: the air is odourless, and the effect is not recognizable to the outside world and parents and is not visible in the behaviour. Some even think that because it is air, that the substance is not retained in the body.

Nitrous oxide use was facilitated, especially among young people, by its low price, apparent safety, easy access and legal status. The hedonistic clip culture, where partying youngsters use in a carefree manner, has also contributed to the popularity. The advent of large kilo tanks, provided by ambitious 24/7 delivery services, has been a dramatic game-changer. From that moment on, due to easier tapping of the gas, the rapidly increasing habituation, and the possibility to inhale nitrous oxide non-stop, repeated dosing seems unlimited. A considerable group of unsuspecting users gradually inhale themselves longer and deeper into an unknown world without a bottom, ceiling, and perspective. Some of them increasingly get lost in hypnotically repetitive and long-lasting nitrous oxide sessions starting from once a week, to a few times a week to sometimes every day. Compared to other substances, nitrous oxide is different in that it can be used for hours and sometimes even days (binge) and the user is not aware when he/she has to stop.

### 4.2. Low Knowledge and Risk Perception

The second question of the investigation referred to the risk perception of (frequent) nitrous oxide use and to what extent this affected their level of use. Initially, the respondents did not classify nitrous oxide as a dangerous substance because of its “innocent” nature, notably because it was not addictive, as confirmed by experts. In addition, nitrous oxide is offered everywhere. Moreover, this group of substance users feels little need to inform themselves and to share their knowledge about the effects, dosages and potential health risks. Especially young people with an intellectual disability encounter difficulties in understanding the information about the risks of nitrous oxide as disseminated by drug prevention organisations. In contrast, clips of artists surrounded by hip-swaying girls with balloons and cool guys who give extra throttle with balloons behind the wheel are often shared by (street) youngsters. In that genre, videos of car accidents are widely viewed, shared and commented on. Discussions about nitrous oxide on the Internet between good and bad guys are also very popular. The values of street culture sometimes resonate in the behaviour surrounding balloon use. You do not want to perform worse than others and you surpass the others with even more balloons. As such, the risks are sought out rather than avoided. Moreover, young people (i.e., the respondents interviewed) are naive and ignorant about psychoactive substances, because they have little experience with them themselves and have no friends who have such experience and knowledge. Apparently, in the absence of the awareness of health risks a self-regulating mechanism is largely lacking. It may happen that someone, as a good Muslim, has never even smoked tobacco or drunk alcohol, but goes completely off track on nitrous oxide in a short time.

### 4.3. Treatment Demand

The problematic users and former users who were interviewed started using nitrous oxide more often and in increasingly higher doses for various reasons. They can be roughly stratified into those with a link to street culture and (sometimes) with contacts in youth and social work in the field, and those with no or weak ties with street culture and have entered the labour market after having finished their education (often intermediate vocational education). They immersed themselves in the festivities, could not keep up with the time, or became adrift (sometimes out of boredom). Some encountered psychological stress (unemployed or high work pressure and the urge to perform) or suffered from an unstable home situation, a lack of recognition and appreciation, or Post Traumatic Stress Disorder (PTSD).

Nitrous oxide provided temporary relief for everyone, with the result that the problems de facto became worse as they began to use more nitrous oxide. In time, this led to various (serious) physical complaints, psychological disorders and social exclusion because of excessive use. Respondents (especially in their abuse phase) experienced many barriers to talk about personal thoughts that bother or frustrate them as well as problems that arise as a result of abuse. They tend to downplay their plight and do not like to express their problems: not to their parents, preferably not to friends, and certainly not to their “street friends”. Often young people say that their pride and honour are at stake and they prefer to keep their problems to themselves. The sense of failure, including that someone’s life is not living up to their parents’ expectations, is widespread.

The continuous switching between different worlds (home, school and circle of friends) takes a lot of energy. Young people often find it difficult to talk about feelings and emotions. The tension of identity, performance and social control between their own community and Dutch society is also reflected in the current results. During the period of their heavy use, the respondents avoided (seeking) help, and/or denied that they were suffering from (serious) emotional problems. In the context of problematic nitrous oxide use (in a taboo culture), it is conceivable that some—on the authority of their environment (including parents, friends)—first sought refuge with an imam or traditional healer. Besides, young people say that they do not have sufficient confidence in assistance by native Dutch medical caregivers or mistrust their help. In addition, they are unfamiliar with them or do not know how to contact them.

### 4.4. Limitations of The Study

Mainly due to difficult access to problematic nitrous oxide users outside of institutional settings and the heavy impediment by feelings of shame among Moroccan youngsters who use nitrous oxide in a problematic way, the sample may not be fully representative and thus does not reflect a statistically random sample. All subjects were recruited from the Dutch–Moroccan community, implying that the results are not representative for Dutch society.

### 4.5. Epilogue

To date, virtually no research has been conducted into the development of problematic nitrous oxide use. The interviews with Moroccan–Dutch youngsters yielded a wealth of information about their environment (including home, school, friends, free time), their views on alcohol and drug use in general, and nitrous oxide use in particular, their drives to use more and more nitrous oxide and how their environment frames substance use in the religious taboo culture.

In collaboration with Reade, Jellinek and the youth work in Amsterdam, New-West, we had the unique opportunity to interview this group of young people (with their permission) in depth. Their personal stories provided a sharper and more nuanced picture of the underlying cultural and social mechanisms related to (problematic) drug use in Moroccan–Dutch youth culture, including their treatment demand. The content of the interviews will be used to fine-tune a targeted reduction strategy for problematic nitrous oxide use in this group. Presumably, the current results also provide an opening for those (parents, friends, professionals, etc.) who are confronted with such problematic users. Anyone who dares to compassionately enter into a conversation with young people will hopefully gain a better understanding as to why some users still see their nitrous oxide tank as their only “friend”.

## Figures and Tables

**Table 1 ijerph-18-05574-t001:** List of topics addressed.

Key Issue	Example
User profile	Starting age, dose, frequency, pathways of progression to problematic use, drug expenditures
Social environment	Upbringing, religion, social control, family, friends, education, work, going out and their expectations for the future
Risk factors	Awareness and experiences of adverse effects, and response to adverse effects
Social and financial problems	Debt accumulation, deterioration of social interactions
Treatment	Counselling, determinants of seeking help or not (shame, sense of urgency, environment)

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
