# Peer review of "Problematic Use of Nitrous Oxide by Young Moroccan–Dutch Adults"

_ijerph, 2021, doi:10.3390/ijerph18115574_

Round 1
Reviewer 1 Report
Re: Peer-Review Report
Dear Authors,
Manuscript ID: ijerph-1207600
Type of manuscript: Article
Title: Problematic use of nitrous oxide by young Moroccan Dutch adults
I have communicated with the journal's editor and recommend accepting your manuscript pending minor revisions (corrections to minor methodological errors and text editing) while moderate English proofreading is required.
Based on critical analysis of the full-text article, the manuscript is original and addresses a novel problem within an ethnic minority, the young Moroccan Dutch adults, within the Dutch community. Although the study is strictly qualitative, it was purposeful and orientating for the reader's attention towards a critical societal issue connected with substance use and misuse.
The background and discussion sections were excellent and conveyed a vital message concerning the necessity of addressing the problematic use of nitrous oxide by young Moroccan Dutch adults via social and medico-legal management plans that may require legislating novel laws incriminating this malicious "pest" of the community.
Further, I am attaching the full-text manuscript in PDF file format, which I highlighted and commented. The authors should comply with all notes embedded in the full-text article, including the remarks printed below
- The title of the study is excellent and reflective for the holistic research and its results
- The abstract is good and comprehensive. However, it lacks reporting descriptive parameters or and numerical results. My first impression after reading the abstract is that the study represents strict qualitative research. If so, the authors should report this aspect early within the abstract.
- The authors need to implement using more appropriate keywords based on Medical Subject Headings (MeSH). Please, check this website by the National Library of Medicine, available at https://www.ncbi.nlm.nih.gov/mesh/
- The introduction section is good but somehow length. The authors need to cut any redundancies and keep this section focused and compact.
- Line-33, the authors need to report a reference for each sentence. Please, check this aspect throughout the full-text manuscript.
- Line-49, the authors should define all abbreviations when they appear first within the text.
- Line-71 to 75, within this paragraph, the authors presented an excellent argument concerning the background and the rationale for conducting their study.
- Line-306, it should be written: "the Holy Quran". Please, double-check the spelling using a well-known dictionary.
- Line-559 to 561, the authors must delete this statement as some readers and critics from the Islamic faith may not tolerate it. Besides, this statement does not add more credibility to the discussion presented by the authors.
- Line-667, the discussion section is good. The authors can benefit by citing these two reference materials:
- Al-Imam A. Monitoring and Analysis of Novel Psychoactive Substances in Trends Databases, Surface Web and the Deep Web, with Special Interest and Geo-Mapping of the Middle East. info:eu-repo/semantics/masterThesis [dissertation on the Internet]. United Kingdom: University of Hertfordshire; 2017. DOI: 10.13140/RG.2.2.27636.24961.
- Al-Imam A, Motyka MA. On the Necessity for Paradigm Shift in Psychoactive Substances Research: The Implementation of Machine Learning and Artificial Intelligence. Alcoholism and Drug Addiction/Alkoholizm i Narkomania. 2019; 32(3): 1-6.
- Line-788, most of the reference materials are suitable and up to date. The authors also need to follow the MDPI journal's guidelines for in-text and bibliographic citations. Please, check at https://www.mdpi.com/authors/references

Author Response
Reviewer 1
I have communicated with the journal's editor and recommend accepting your manuscript pending minor revisions (corrections to minor methodological errors and text editing) while moderate English proofreading is required.
Answer by authors: Thank you for your constructive and critical remarks. The revised text has been corrected by a native speaker.
Based on critical analysis of the full-text article, the manuscript is original and addresses a novel problem within an ethnic minority, the young Moroccan Dutch adults, within the Dutch community. Although the study is strictly qualitative, it was purposeful and orientating for the reader's attention towards a critical societal issue connected with substance use and misuse.
The background and discussion sections were excellent and conveyed a vital message concerning the necessity of addressing the problematic use of nitrous oxide by young Moroccan Dutch adults via social and medico-legal management plans that may require legislating novel laws incriminating this malicious "pest" of the community.
Further, I am attaching the full-text manuscript in PDF file format, which I highlighted and commented. The authors should comply with all notes embedded in the full-text article, including the remarks printed below
The title of the study is excellent and reflective for the holistic research and its results.
Answer by authors: Thank you for your constructive and critical remarks.
- The abstract is good and comprehensive. However, it lacks reporting descriptive parameters or and numerical results. My first impression after reading the abstract is that the study represents strict qualitative research. If so, the authors should report this aspect early within the abstract. Answer by authors: Good point. We have added in the abstract that the study is a qualitative explorative study. In paragraph 1.3 and at the end of the introduction this term was inserted, as well.
- The authors need to implement using more appropriate keywords based on Medical Subject Headings (MeSH). Please, check this website by the National Library of Medicine, available at https://www.ncbi.nlm.nih.gov/mesh/
Answer by authors: we have added some appropriate key words. E.g.: nitrous oxide, adverse effects, recreational use, drug dependence, migrants.
- The introduction section is good but somehow length. The authors need to cut any redundancies and keep this section focused and compact.
Answer by authors: We have revised the introduction by deleting less relevant information to increase the focus. A reduction on word count of 33% was achieved.
- Line-33, the authors need to report a reference for each sentence. Please, check this aspect throughout the full-text manuscript.
Answer by authors: Line 33 this is commonly known. Reference is not given to limit the number of references. The other parts of the MS have been checked for missing references.
- Line-49, the authors should define all abbreviations when they appear first within the text. Answer by authors: We agree. MBO/HBO has been replaced by “post-secondary vocational education”. Other abbreviations have been spelled out when appearing first within text.
Line-71 to 75, within this paragraph, the authors presented an excellent argument concerning the background and the rationale for conducting their study.
- Line-306, it should be written: "the Holy Quran". Please, double-check the spelling using a well-known dictionary.
Answer by authors: We agree. Has been rephrased accordingly.
- Line-559 to 561, the authors must delete this statement as some readers and critics from the Islamic faith may not tolerate it. Besides, this statement does not add more credibility to the discussion presented by the authors.
Answer by authors: We agree. Not very relevant and could be deleted (as done).
- Line-667, the discussion section is good. The authors can benefit by citing these two reference materials:
- Al-Imam A. Monitoring and Analysis of Novel Psychoactive Substances in Trends Databases, Surface Web and the Deep Web, with Special Interest and Geo-Mapping of the Middle East. info:eu-repo/semantics/masterThesis [dissertation on the Internet]. United Kingdom: University of Hertfordshire; 2017. DOI: 10.13140/RG.2.2.27636.24961.
- Al-Imam A, Motyka MA. On the Necessity for Paradigm Shift in Psychoactive Substances Research: The Implementation of Machine Learning and Artificial Intelligence. Alcoholism and Drug Addiction/Alkoholizm i Narkomania. 2019; 32(3): 1-6.
Answer by authors: Thank you. However, we do not see the relevancy of these two references within the scope of the current paper i.e. beyond scope.
- Line-788, most of the reference materials are suitable and up to date. The authors also need to follow the MDPI journal's guidelines for in-text and bibliographic citations. Please, check at https://www.mdpi.com/authors/references.
Answer by authors: Thank you. We have checked.
Reviewer 2 Report
Dear authors,
The topic is original and interesting. However I believe there are several limitations with the article that need to be addressed:
You need to proofread it carefully - there were several mistakes (e.g. line 147) and poorly constructed sentences and you repeat a title (subheading 1.2). Also be careful with the vocabulary used (e.g. shady to describe a behaviour). All the acronyms need to be written down before used (e.g. MBO). Review the English used throughout the text;
Subheading 1.4 - the text here should be clearer. You should summarise the evidence in a clear way that makes it a fluid read. Also was not clear for me what you intended with the comparison with Black Americans in line 184;
Methods - I believe this section needs a lot of improvement: was not clear how you chose the participants, how the questions for the in-depth interview were built; what was the difference for your research question and for your interview (you mention the time difference) between the participants who were in treatment and those that were not, for example;
The results are long. I would suggest summarising what is relevant and present some results with appropriate meaningful citations;
Subheading 3.4 - there is a mix between results and theory here;
Discussion - I suggest you address clearly the limitations of this study, what do these results mean and more importantly, the implications for practice and further research.
The article is also very long and could benefit eventually from making it more fluid with the key ideas more organised.
Author Response
Reviewer 2
- The topic is original and interesting.
Answer by authors: Thank you for your constructive and critical remarks.
However I believe there are several limitations with the article that need to be addressed:
- You need to proofread it carefully - there were several mistakes (e.g. line 147)
Answer by authors: Correct remark. This line has been revised.
- and poorly constructed sentences and you repeat a title (subheading 1.2).
Answer by authors: Correct remark. This line has been corrected.
- Also be careful with the vocabulary used (e.g. shady to describe a behaviour).
Answer by authors: Correct remark. Rephrased by “express themselves vaguely about”
- All the acronyms need to be written down before used (e.g. MBO).
Answer by authors: Correct remark. Abbreviations have been spelled out when appearing first within text.
- Review the English used throughout the text;
Answer by authors: the revised text has been corrected by a native speaker.
- Subheading 1.4 - the text here should be clearer. You should summarise the evidence in a clear way that makes it a fluid read.
Answer by authors: this section has been largely restructured and rephrased, making it much clearer to the reader.
- Also was not clear for me what you intended with the comparison with Black Americans in line 184;
Answer by authors: Correct remark. This line has been deleted.
- Methods - I believe this section needs a lot of improvement: was not clear how you chose the participants, how the questions for the in-depth interview were built; what was the difference for your research question and for your interview (you mention the time difference) between the participants who were in treatment and those that were not, for example;
Answer by authors: Fair point. Obviously, our recruitment procedure did not produce a statistically random sample, which was mainly due to heavy impediment to get access to problematic nitrous oxide users outside of institutional settings, and by feelings of shame among this group of problematic nitrous oxide users. In the revised Methods section we have further outlined the procedure which includes now a table with the most relevant key issues addressed.
- The results are long. I would suggest summarising what is relevant and present some results with appropriate meaningful citations;
Answer by authors: We do not agree. This is a qualitative study which allows to describe the main points expressed during the interviews. Redundant phrases have been deleted where possible.
- Subheading 3.4 - there is a mix between results and theory here;
Answer by authors: Good point. Indeed, occasionally results and theory have been mixed. We have therefore moved the paragraph about set and setting (Zinberg model) in a more concise version to the Discussion section.
- Discussion - I suggest you address clearly the limitations of this study, what do these results mean and more importantly, the implications for practice and further research.
A section with limitations of the study has been inserted. The implications for practice and further research has been described now in the Discussion. The following lines has been inserted: “The content of the interviews will be used to fine-tune a targeted reduction strategy for problematic nitrous oxide use in this group.”
- The article is also very long and could benefit eventually from making it more fluid with the key ideas more organised.
Answer by authors: Where possible sections have been shortened and redundant text removed. In addition, we have inserted an organised table (Table 1) with the key issues addressed.

Reviewer 3 Report
The article deals with the up-to-date social topic regarding abuse and social circumstances of nitrous oxide in Moroccoan-Dutch young adults. Text works with original idea however there are several key points to be improved.
Structure lacks of arranged form. Many topic are closed at some point and is discussed further on and due its repetition it takes a lot of space. Text is sometimes confusing and has to be read several times to get the main information. Citation is used a lot of times and review table might be better.
English is used in a perfect level but few points need a correction. From the line 78 there are missed brackets, characters or even unexplained abbreviations.
Though the introduction is well structualized and well thought, the results and discussion address too many subtopics to handle and therefore the key research message can not be easily seen.
Recommendation would be to amend the whole text and focus on few key points only or to consider other journal for publication.
Author Response
Reviewer 3
The article deals with the up-to-date social topic regarding abuse and social circumstances of nitrous oxide in Moroccan-Dutch young adults. Text works with original idea however there are several key points to be improved.
Answer by authors: Thank you for your constructive and critical remarks.
Structure lacks of arranged form. Many topic are closed at some point and is discussed further on and due its repetition it takes a lot of space. Text is sometimes confusing and has to be read several times to get the main information. Citation is used a lot of times and review table might be better.
Answer by authors: Where possible sections have been shortened and redundant text removed. In addition, parts have been restructured and we have inserted an organised table (Table 1) with the key issues addressed.
English is used in a perfect level but few points need a correction. From the line 78 there are missed brackets, characters or even unexplained abbreviations.
Answer by authors: These failures have been checked and corrected in the revised MS.
Though the introduction is well structualized and well thought, the results and discussion address too many subtopics to handle and therefore the key research message cannot be easily seen.
Answer by authors: The introduction has been restructured and is now limited to background information related to the issues raised during the interviews. We agree that number of topics raised is high and the issues very diverse, and therefore confusing. To increase visibility of the core issues, we have inserted a table describing the key issues.
Recommendation would be to amend the whole text and focus on few key points only or to consider other journal for publication.
Round 2
Reviewer 2 Report
Dear authors,
I believe you did a really good job at improving the manuscript. I would just suggest proofreading it carefully and making sure everything is in place (for example I did not see a reference for Table 1 in the text).
Overall, great effort.
Reviewer 3 Report
Article describes severe neurological symptoms in Amsterdam care centres as an abuse of nitrous oxide in Moroccan-Dutch young adults. Authors chose current topic dealing with the socioeconomic issue in European-imigrant environment suffering from identity struggle.
Article is well structuralized, has clearly set aims, questions and results. Individual sub-topics are organized and evident. Recommended for publishing.